# Variable bites and dynamic populations; new insights in *Leishmania* transmission

**Samuel Carmichael**[1], **Ben Powell**[1], **Thomas Hoare**[2], **Pegine B. Walrad**[2]\*, **Jonathan W. Pitchford**[1,2]\*

**1** Department of Mathematics, University of York, York, UK, **2** Department of Biology, University of York, York, UK

\* pegine.walrad@york.ac.uk (PBW); jon.pitchford@york.ac.uk (JWP)

## Abstract

Leishmaniasis is a neglected tropical disease which kills an estimated 50,000 people each year, with its deadly impact confined mainly to lower to middle income countries. *Leishmania* parasites are transmitted to human hosts by sand fly vectors during blood feeding. Recent experimental work shows that transmission is modulated by the patchy landscape of infection in the host's skin, and the parasite population dynamics within the vector. Here we assimilate these new findings into a simple probabilistic model for disease transmission which replicates recent experimental results, and assesses their relative importance. The results of subsequent simulations, describing random parasite uptake and dynamics across multiple blood meals, show that skin heterogeneity is important for transmission by short-lived flies, but that for longer-lived flies with multiple bites the population dynamics within the vector dominate transmission probability. Our results indicate that efforts to reduce fly lifespan beneath a threshold of around two weeks may be especially helpful in reducing disease transmission.

## Author summary

Two recent discoveries hold particularly important ramifications for *Leishmania* transmission. First, parasites are heterogeneously distributed within the skin of an infected host. Second, the discovery of a new lifecycle stage known as the retroleptomonad promastigote changes the within-vector parasite dynamics. It is not yet known how these newly identified factors may interact to influence transmission. In this study, we design a tractable model for parasite population dynamics in the sand fly vector that consolidates these new results into a single system. We first demonstrate that our model can replicate established experimental results. We then interrogate this model, both analytically and numerically, to draw conclusions about *Leishmania* transmission in an ecological and epidemiological context. We conclude that the relative importance of the two focal factors depends critically on sand fly lifespan. In short-lived sand flies the heterogeneity in the number of parasites initially taken up by a sand fly is typically the crucial factor in *Leishmania* transmission, whereas for longer-lived sand flies the retroleptomonad lifecycle stage is likely to drive transmission. In a practical context these results suggest that

**Data Availability Statement:** All relevant code necessary to produce the required simulation output is included within the manuscript and its Supporting information files.

**Funding:** SC received funding as part of an EPSRC (epsrc.ukri.org) funded PhD within the Department of Mathematics, University of York. JWP and PBW received funding from the Department of Biology, University of York (www.york.ac.uk) to support TH. The funders had no role in study design, data collection and analysis, decision to publish, or preparation of the manuscript.

**Competing interests:** The authors have declared that no competing interests exist.

minimising the expected sand fly lifespan could be an effective strategy to reduce *Leishmania* transmission.

## Introduction

Leishmaniasis is caused by parasites of the *Leishmania* genus. Details of the infection depend on the particular species [1], but all species share the same general vector-borne lifecycle, with distinct and complex life cycle stages in the mammalian host and sand fly vector [2]. *Leishmania* parasites have two main morphological forms. Broadly speaking, amastigotes (ovoid, non-flagellated) dominate the mammalian stage of the lifecycle. Promastigotes (larger, flagellated) are found in the vector, and are divided into multiple developmental subclasses [3, 4].

Sand flies in natural settings are often opportunistic feeders, capable of feeding on a variety of mammalian and avian species [5, 6]. Mature female sand flies require a blood meal during each oviposition cycle. When an uninfected female sand fly bites an infected mammal, it ingests amastigote-infected macrophages from the host's skin or blood [7]. Within the first few days, amastigotes differentiate into procyclic promastigotes, which are resistant to the digestive enzymes of the sand fly midgut [2]. Procyclics then exponentially replicate before differentiating into nectomonad promastigotes [3]. Nectomonads are able to migrate towards the thoracic midgut [2] and bind to the midgut epithelium [8] where they differentiate into leptomonad promastigotes [3]. Leptomonads are the second replicative stage, and migrate through the thoracic midgut to the stomodeal valve [3] where these differentiate into metacyclic promastigotes, the human-infectious stage. Metacyclics have a short cell body and long flagellum to enhance motility [3], and can be transmitted to a new host where they infect host macrophages via phagocytosis. (The infection dynamics in the host are similarly complex [9, 10], but are not relevant to this investigation which focuses on transmission potential from vector to host.) Two recent key findings concerning details of *Leishmania* biology offer new insights into the possibility of understanding, and possibly controlling, the spread of the disease. They are described below.

**Patchy landscape of infection in the host** Transmission from host to vector occurs when a sand fly consumes a blood meal from an infected host. Doehl *et al.* [7] examined amastigote *Leishmania donovani* infections in immunodeficient mice. By evaluating the correlation of the sand fly parasite burden with multiple measures of host parasite burden, they showed first that the parasite load in mammalian host skin, rather than blood, is the major determinant of successful sand fly infection. They further found that skin parasite burden is highly variable within and between mammalian hosts and developed a modelling approach to investigate the consequences of this patchiness. For a host with a low mean parasite burden, a patchy skin landscape enhanced outward transmission (although the overall probability of successful transmission remained low), whereas for a host with a high parasite burden a homogenous distribution favoured transmission.

**Retroleptomonads** A new lifecycle stage was identified by Serafim *et al.* [11], the retroleptomonad promastigote [11]. When a sand fly with a mature (metacyclic enriched) infection takes another blood meal, the metacyclic stage can de-differentiate into a leptomonad-like stage, termed the retroleptomonad. These replicate for 3-4 days before differentiating back into metacyclics [11]. This serves to greatly amplify the parasite load prior to the next bite (4.5 fold increase in the number of metacyclics 18 days post infection in comparison to a sand fly that has fed only once) and thus increases the probability of disease transmission [11], a finding confirmed experimentally under laboratory conditions.

Doehl *et al.* [7] observed that often the sand flies would only carry a relatively small infection after a single feed, suggesting that perhaps sand flies may only be expected to infect once they had taken 2 previous bites (and thus had their infection amplified via the the retroleptomonad stage [11]), but the correlation between these two mechanisms has not yet been fully explored.

The objective of the work presented here is to build a mathematical model to incorporate these new findings and assess the impact upon *Leishmania* transmission. A simple differential equation model, parameterised by data from [3], was developed to describe the population dynamics of nectomonad, leptomonad and metacyclic promastigote stages within the vector (Model A). This model was then refined by the addition of the retroleptomonad lifecycle stage, using data and observations from [11] (Model B). These models of population dynamics within the sand fly provide a framework for a series of stochastic simulations which describe the random processes of feeding and parasite ingestion across multiple blood meals. Such simulations allow the consequences of changes in disease prevalence at the epidemiological scale and the thresholds of disease transmission to be quantifiably predicted.

# 1 Model details

## 1.1 Modelling approach

The modelling strategy is summarised in Fig 1. First, we develop a simple, algebraically tractable and computationally efficient model for parasite population dynamics within a single infected sand fly, and then parameterise this model according to the available information. This model then forms a key ingredient in a series of larger stochastic simulations intended to extract useful details about the transmission of *Leishmania*.

In order to create a tractable model, several key assumptions are made. In addition to those represented in Fig 1, we also assume that differentiation between parasite life cycle stages occurs at 100% efficiency and that there is a single globally applied sand fly carrying capacity of *Leishmania* parasites.

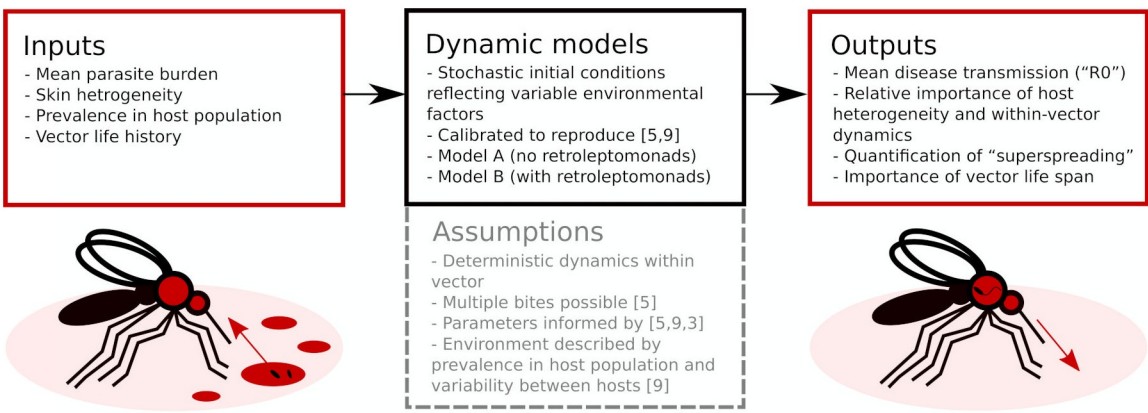

**Fig 1. Flowchart overview of the modelling approach.** Two dynamic models, calibrated to replicate prior results, evaluate parasite population dynamics in the sand fly vector. These can be used as part of larger simulations to obtain insights into *Leishmania* transmission.

## 1.2 Model definitions

Model A describes the dynamics of Nectomonads ($N$), Leptomonads ($L$) and Metacyclics ($M$) using a simple set of near-linear ordinary differential equations (ODEs),

$$\frac{dN}{dt} = -\alpha N \tag{1}$$

$$\frac{dL}{dt} = \alpha N + rL\left(1 - \frac{N + L + M}{C}\right) - sL \tag{2}$$

$$\frac{dM}{dt} = sL - uM \tag{3}$$

The assumptions are biologically parsimonious: $N$ differentiate into $L$ at rate $\alpha$, $L$ replicate at rate $r$ (limited by a carrying capacity $C$) and differentiate to $M$ at rate $s$, and $M$ are also subject to mortality at rate $u$.

Model B extends Model A to incorporate the dynamics of the Retroleptomonads ($R$) [11] using two sets of near-linear ODEs. Under standard conditions 'normal mode' is used,

$$\frac{dN}{dt} = -\alpha N \tag{4}$$

$$\frac{dL}{dt} = \alpha N + rL\left(1 - \frac{N + L + M + R}{C}\right) - sL \tag{5}$$

$$\frac{dM}{dt} = sL + vR - uM \tag{6}$$

$$\frac{dR}{dt} = qR\left(1 - \frac{N + L + M + R}{C}\right) - vR \tag{7}$$

In addition to the original assumptions, it is assumed that any existing $R$ differentiate to $M$ at rate $v$ and replicate at rate $q$ limited by carrying capacity $C$. For a four-day period after subsequent bites 'dedifferentiation mode' is used,

$$\frac{dM}{dt} = sL - gM - uM \tag{8}$$

$$\frac{dR}{dt} = qR\left(1 - \frac{N + L + M + R}{C}\right) + gM \tag{9}$$

Now, $M$ dedifferentiate to $R$ at rate $g$ and $R$ no longer differentiate to $M$.

Parameterisation of Model A was performed using data obtained from Rogers *et al* [3] (see S1 Method) but due to a lack of suitable data, it was not possible to perform similar parameter fitting for the new parameters in Model B.

Table 1 includes a summary of the default parameter values chosen.

For an implementation of the above models see Supplementary S1 Code.

**Table 1. Table of default model parameter values.**

| Parameter | Name | Default Value | Units | Source |
|-----------|------|---------------|-------|--------|
| $\alpha$ | Nectomonad differentiation rate | 1.52 | $d^{-1}$ | [A] |
| $r$ | Leptomonad replication rate | 1.45 | $d^{-1}$ | [A] |
| $s$ | Leptomonad differentiation rate | 1.65 | $d^{-1}$ | [A] |
| $u$ | Metacyclic decline rate | 1.61 | $d^{-1}$ | [A] |
| $C$ | Carrying capacity | $2 * 10^6$ | *individuals* | [B] |
| $v$ | Retroleptomonad differentiation rate | 4.0 | $d^{-1}$ | [B] |
| $q$ | Retroleptomonad replication rate | 3.5 | $d^{-1}$ | [B] |
| $g$ | Metacyclic dedifferentiation rate | 4.0 | $d^{-1}$ | [B] |

All parameters and their default values. [A]: Values are derived from parameterisation based on data from Rogers *et al*. [3], see S1 Method. [B]: Parameter estimates chosen to be consistent with population data from Serafim *et al*. [11].

## Results

### 1.2.1 Replicating experimental results on sand fly feeding schedules and mammalian infection heterogeneity

In order to verify that our retroleptomonad-inclusive Model B is capable of replicating the experimental results observed by Serafim *et al*. [11], we ran a set of 20,000 Monte Carlo simulations designed to imitate their experimental setup. In this scenario, all flies take a bite at day 0 from an infected host. Half the flies take an additional bite at day 12 from an uninfected host, the other half take no subsequent bites. We fix the mean skin parasite burden to $2 \times 10^6$ and let $k = 2$ to mimic the blood source used by Serafim *et al*. After the initial bite, we take up a number of amastigotes according to the methods in S2 Method. In this example, the initial number of nectomonads $N_0$ has mean $\mu$ and variance $\sigma^2$:

$$\mu = 9,600 \qquad \sigma^2 = 46,108,800$$

Of particular interest are the numbers of metacyclics and retroleptomonads present in each fly throughout their adult lifespan. Fig 2A compares the numbers of metacyclics and retroleptomonads at each day sampled by Serafim *et al*.

Fig 2A reflects the qualitative dynamics observed in the experiments of Serafim *et al*. We observe a similar reduction in the number of metacyclics immediately after the bite at day 12 and a corresponding increase in the number of retroleptomonads over the same time period. Similar behaviour can be observed for the proportions of metacyclics and retroleptomonads (S1 Fig), and this behaviour is sufficiently robust to be observed even with parameter randomisation (S2 Fig).

We also wish to verify that our model can describe the role of heterogeneity in the skin parasite distribution as reported by Doehl *et al* [7]. To do so, we ran sets of 1000 Monte Carlo simulations for parameter combinations corresponding to mice 10-18 as calculated by Doehl *et al* (S1 Table). Each simulated fly fed on an infected host at $t = 0$. We then sampled the number of metacyclics in each fly after 7 days. Based on the work of Sadlova *et al*. [12], we consider a sand fly to be infectious if 500 metacyclics are present at day 7 post-infection. This is a distinct, but similar, approach to that of Doehl *et al*. [7] Whereas Doehl *et al*. predicted the number of flies with mature infections based upon amastigote uptake, we evaluate this number directly using a comparable threshold. Fig 2B compares the number of infectious sand flies for each mouse.

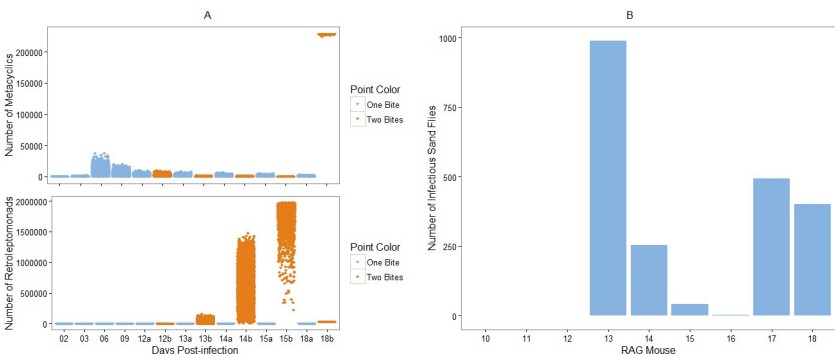

**Fig 2. Replicating the results of [7] and [11].** A) Comparison of the numbers of metacyclics (top) and retroleptomonads (bottom) at specific days throughout the lifespan of the simulated flies. Blue represents flies that bite only at day 0, orange represents flies that take a subsequent blood meal at day 12. The two categories are combined prior to day 12. B) Number of simulated sand flies considered infectious at 7 days post-infection for RAG mice 10-18, parameterised according to Doehl *et al.* (see S1 Table.).

We observe that heavily infected mice, such as mouse 13, result in a large proportion, if not all, of the sand flies being mammalian-infectious at day 7 post-infection (S1 Table). Relatively smaller infections, such as those of mice 10 and 16, typically lead to negligibly-infectious sand flies. This matches the observations made by Doehl *et al* [7] and verifies that our model successfully captures the relationship between outward transmission potential and skin patchiness.

## 1.3 Analytic results

In this section we provide analytically-derived properties and consequences of simplified versions of our models. These serve to reinforce and validate the numerically derived behaviours discussed in Section 1.4 and to highlight the key processes driving transmission. In particular, we present expressions bounding implied disease transmission probabilities in a range of hypothetical scenarios.

In order to render it analytically tractable, it is necessary to make two simplifications to our model. Explicitly, we assume that 1) blood meals only occur at specific predetermined times, rather than at random gamma-distributed times as in the full model, 2) no sand fly mortality occurs during our simulations. This simplifies the probabilistic model such that the only random variables affecting the parasite transmission events are the initial number of parasites present in the sand fly, and the presence or absence of a second blood meal.

More specifically, we restrict our attention to scenarios in which a sand fly takes either two or three blood meals over a period of 12 days. In all scenarios let $N_0$ be the number of necto-monads present in the sand fly 4 days post-blood meal. We choose $t = 0$ such that each sand fly initially carries $N_0$ nectomonads. We also assume that the fly feeds on an uninfected host at time $t = 12$, when it deposits $M_{12}$ parasites in the metacyclic life cycle stage. $N_0$ is considered a random variable. $M_{12}$ is considered a deterministic function of $N_0$, and so inherits probabilistic behaviour from this random variable. A transmission event is associated with the sand fly depositing a number of parasites ($M_{12}$) exceeding a threshold $T$. Thus transmission is also a random variable inheriting probabilistic behaviours from $N_0$.

The scenarios we consider differ in terms of the occurrence of an additional blood meal from an uninfected host at time $t = 6$. In our model, this 2nd ingested blood meal triggers differentiation to the retroleptomonad lifecycle stage, associated replication and re-differentiation

back to metacyclic stage, impacting the number of metacyclics that can be deposited at time $t = 12$.

Given that there are blood meals only at times 0 and 12, the structure of the model described in Section 1 is such that $M_{12}$ is proportional to $N_0$ i.e.

$$M_{12} = C_2 N_0 \tag{10}$$

where $C_2$ is a constant derived by solving the system of equations in Section 1. It is implicitly a function of the model's differentiation rate parameters and the time elapsed between blood meals.

If an additional blood meal at time $t = 6$ does occur, a different set of equations that involve the retroleptomonads is used to determine the resulting number of metacyclics at time $t = 12$. $M_{12}$ is now determined by $N_0$ and a correspondingly different multiplicative constant

$$M_{12} = C_3 N_0 \tag{11}$$

Expressions (10) and (11) can be combined to give

$$M_{12} = C_3 N_0 \mathbf{1}_B + C_2 N_0 (1 - \mathbf{1}_B) \tag{12}$$

where $\mathbf{1}_B$ is an indicator function taking value one when the $t = 6$ blood meal occurs, and zero otherwise.

We can now, for instance, consider the expectation of $M_{12}$

$$\begin{aligned}
\mathbb{E}(M_{12}) &= C_3 \mathbb{E}(N_0) \mathbb{E}(\mathbf{1}_B) + C_2 \mathbb{E}(N_0)(1 - \mathbb{E}(\mathbf{1}_B)) \\
&= [C_2 + (C_3 - C_2)\mathbb{E}(\mathbf{1}_B)]\mathbb{E}(N_0)
\end{aligned} \tag{13}$$

which follows on the assumption that $\mathbf{1}_B$ and $N_0$ are considered probabilistically independent. Note that $\rho := \mathbb{E}(\mathbf{1}_B)$ is the probability that the blood meal bite takes place.

Eq (12) can also be used to produce an expression for the transmission probability at time $t = 12$

$$\begin{aligned}
P(\text{Transmission}) &= P(M_{12} \geq T) \\
&= P(M_{12} \geq T \mid \text{second bite})P(\text{second bite}) \\
&\quad + P(M_{12} \geq T \mid \text{no second bite})P(\text{no second bite}) \\
&= P(N_0 \geq T/C_3)\mathbb{E}(\mathbf{1}_B) + P(N_0 \geq T/C_2)(1 - \mathbb{E}(\mathbf{1}_B))
\end{aligned} \tag{14}$$

We will use Eq (14) to express how the variability in $N_0$, which was the subject of interest in Doehl *et al.* [7], and the variability in the blood meal availability, which was the subject of interest in Serafim *et al.* [11], both contribute to the probability of disease transmission.

To help progress our arguments here we appeal to Chebyshev's inequality, which tells us that a random variable takes values close to its expectation with high probability, more precisely it says that the probability of the random variable being further than $k > 0$ standard deviations from the expectation is smaller that $k^{-2}$ i.e.

$$P(|X - \mathbb{E}(X)| \geq k\sqrt{\mathrm{var}(X)}) \leq 1/k^2 \tag{15}$$

or equivalently

$$P(|X - \mathbb{E}(X)| \geq k) \leq \mathrm{var}(X)/|k|_+^2 \tag{16}$$

where we have introduced the rectifier function

$$|k|_+ = \begin{cases} k & k > 0 \\ 0 & k \leq 0 \end{cases} \tag{17}$$

in order to accommodate negative $k$.

In the case when there is no bite at time $t = 6$ Chebyshev's inequality allows us to put an upper bound on the transmission probability

$$
\begin{aligned}
P[\text{Transmission} \mid \text{no second bite}] &= P[M_{12} \geq T \mid \text{no second bite}] \\
&= P[C_2 N_0 \geq T] \\
&= P[N_0 - \mathbb{E}(N_0) \geq T/C_2 - \mathbb{E}(N_0)] \\
&\leq P[|N_0 - \mathbb{E}(N_0)| \geq T/C_2 - \mathbb{E}(N_0)] \\
&\leq \text{var}\,(N_0)/|T/C_2 - \mathbb{E}(N_0)|_+^2
\end{aligned} \tag{18}
$$

Such an upper bound is useful because it suggests ways the transmission probability can, in principle at least, be forced down. We could, for example, force down the variance of the number of parasites ingested at time $t = 0$. Alternatively, by decreasing the conversion rate from nectomonads at time $t = 0$ to metacyclics at time $t = 12$ we would decrease $C_2$ which also serves to bring down the upper bound.

Considering the average over cases in which the blood meal bite does and does not occur at time $t = 6$, Chebyshev's inequality leads us to an expression of the form

$$
\begin{aligned}
P[\text{Transmission}] &= P[M_{12} \geq T] \\
&\leq \text{var}\,(N_0) \left( \frac{\rho}{|T/C_3 - \mathbb{E}(N_0)|_+^2} + \frac{1-\rho}{|T/C_2 - \mathbb{E}(N_0)|_+^2} \right) \\
&\leq \text{var}\,(N_0) \frac{1}{|T/(\rho C_3 + (1-\rho)C_2) - \mathbb{E}(N_0)|_+^2} \\
&= \text{var}\,(N_0) \frac{1}{|T'/C_2 - \mathbb{E}(N_0)|_+^2}
\end{aligned} \tag{19}
$$

where the second line follows from Jensen's inequality. Since $C_3 > C_2$, the second bite/retro-leptomonad phenomenon effectively leads to a version of Eq (18) in which the transmission threshold has been lowered from $T$ to

$$T' = T \times \frac{1}{1 + \rho(C_3/C_2 - 1)} \tag{20}$$

As well as providing quantitative predictions, this 'equivalent threshold' result is intended to provide another angle from which to interpret the significance of the retroleptomonad reproduction mechanism. Specifically, the retroleptomonads do not negate the capacity for skin heterogeneity to increase metacyclic numbers to transmission-sufficient levels for a subset of flies. Rather, they make these levels easier to attain. We see the effects of skin heterogeneity and the retroleptomonads act together to contribute to disease transmission.

An alternative expression linking the retroleptomonads to the transmission probability follows from assuming that the number of metacyclics derived from retroleptomonads is very large relative to the transmission threshold (i.e. $C_3 N_0 \gg T$). In this case we can consider the

transmission probability, given the blood meal bite at $t = 6$, is close to one

$$P(M_{12}^* \geq T \mid \text{second bite}) \approx 1 \tag{21}$$

Then, using Chebyshev's Inequality we see that

$$
\begin{aligned}
P(M_{12}^* \geq T) \quad &\leq \rho + (1 - \rho) \frac{\mu_{M^*}(1 + \mu_{M^*}/k)}{\left(T/C_2 - \mu_{M^*}\right)^2} \\
&= \rho + (1 - \rho) \frac{\mathrm{var}\,(N_0)}{\left(T/C_2 - \mathbb{E}(N_0)\right)^2}
\end{aligned}
\tag{22}
$$

This bound provides another way to assess the relative influences of key parameters on the probability of transmission. For cases in which the transmission threshold is high relative to the number of metacyclics produced without the retroleptomonads (i.e. $C_2 N_0 \ll T$) and the blood meal bite probability $\rho$ is reasonable large, the rightmost summand in Eq (22) dominates. We then see the transmission probability reduced to the blood meal bite probability. When $\rho$ is very small, however, the variance of $N_0$, and the skin heterogeneity that drives it, becomes important again. In this case it is this heterogeneity that provides each sand fly with the greatest likelihood of depositing a sufficient number of *Leishmania* parasites at time $t = 12$ to cause transmission.

Our simplified model, via Eq (22), re-frames the competing roles of the second blood meal and the skin heterogeneity in a mathematically precise way. The simulations and discussions below do the same at increasing levels of realism, but necessarily decreasing levels of mathematical formalism.

## 1.4 Simulation study

This simplified model is useful because it allows us to make analytical predictions about the behaviour of our system. However such predictions are useful only where their implications can be related to more sophisticated systems. Let us once more consider the full system for both models as originally defined (Model A: Eqs 1-3; Model B: Eqs 4-9). Each sexually mature female fly has a predetermined lifespan drawn from an exponential distribution with a mean and standard deviation of 13 days. These sand flies bite throughout their lives, with inter-bite times drawn from a gamma distribution of mean 6 days, standard deviation $\sqrt{3}$ days and with bite loads as previously defined (S2 Method). We also reinstate a 3-day delay before the emergence of nectomonads and assume that all sand flies are initially uninfected.

We require a suitable metric to assess the infectiousness of *Leishmania* under a variety of $P_B$ and k values. One such metric commonly used in epidemiology is the $R_0$ [13] defined as "the number of secondary infections generated from a single infected individual introduced into a susceptible population" [14]. As we do not explicitly model individual hosts, this measure is unsuitable. Let us instead consider a proxy value: mean sand fly transmission capacity (hereafter referred to as mean $R_0$), defined to be the average number of infections caused by a single sand fly. Though this is not strictly an $R_0$ value, higher mean $R_0$ values imply a higher $R_0$ value for the disease assuming that the number of sand flies biting a given infected host remains unchanged.

We determine that a transmission has occurred at a given bite using either a binary threshold or a smooth 'threshold function'. In the case of the binary threshold, we assume that if the number of metacyclics transferred ($M_T$) exceeds some fixed threshold T, an infection is guaranteed (and if not an infection never occurs). For the smooth 'threshold function', we

assume the chance of infection $P_T$ at a given bite depends on $M_T$ such that:

$$P_T = 0.5(\tanh(0.015(M_T - 200)) + 1) \tag{23}$$

Whilst the binary threshold is easier to relate to our analytical work it is very unlikely to be applicable to a real situation, especially as it disregards any nutritional or genetic variation between potential hosts. Thus, let us consider the smooth threshold function. Corresponding figures for the binary threshold function can be found in the supplementary information, and we observe qualitatively similar behaviour with both the binary and smooth thresholds.

We compare our two models' outputs for a range of different scenarios. Assume that some proportion of hosts is initially infected and that this proportion is fixed with no dependence on time or transmissions. Initially, we will consider two scenarios where our simulated flies bite at random from a population of hosts in which either 100%, or 25%, of hosts are infected (see Fig 3; for further scenarios see S3 Fig. and for the binary threshold equivalent see S4 Fig).

Although the simplest conclusion we can draw from these heatmaps is that introducing retroleptomonads increases our mean $R_0$ value, there are several other notable results. We observe that for Model A there is a peak in the mean $R_0$ value for low skin homogeneity and high mean skin parasite burden for both scenarios. Though our analytic approach does not deal directly with Model A, we could consider Model A to simply be the scenario where flies never take 3 blood meals (and thus where the retroleptomonad lifecycle stage has no significant role in day 12 transmission). In this context, we note that a low skin homogeneity increases the probability of transmission as some flies are able to ingest a sufficient number of parasites to become infectious by the next blood meal. In contrast, more homogeneous skin environments reduce the probability that any individual sand fly would ingest sufficent parasite numbers for strong transmission capacity. These findings support the prediction of Doehl *et al.* [7].

The peak is entirely absent from the corresponding heatmaps for Model B; instead we have a plateau spanning most of the parameter space with a slight decrease in mean $R_0$ for very low

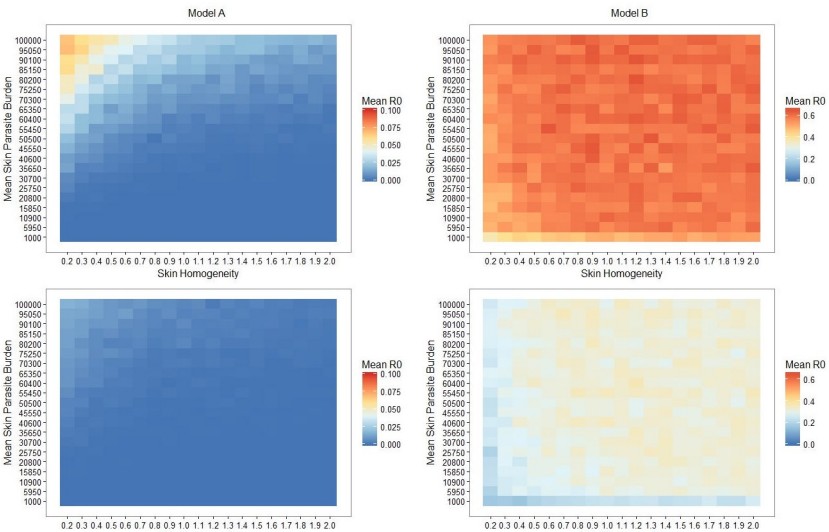

**Fig 3. Retroleptomonad dynamics dominate over skin heterogeneity and result in elevated mean $R_0$ values.**
Heatmaps of the mean $R_0$ for simulated sand flies for both Model A (left half) and B (right half) with 100% (top half) or 25% (bottom half) chance of biting an infected host. Note that each model utilises a different scale for clarity.

k values (i.e. very patchy environments). We note from our analytical section that as $\rho$ (the chance of taking 3 bites) increases, k (skin homogeneity) has a progressively reduced impact. Thus, given that $\rho$ effectively remains constant (and non-zero) regardless of k one might anticipate that the mean $R_0$ would be independent of k. Similarly, considering the magnitude of the amplification of the metacyclics (Fig 2A) it is reasonable to expect that the mean skin parasite burden would be relatively unimportant. This does not hold for very low skin homogeneity and/or parasite burdens, because under these conditions it is possible that the sand fly may fail to be initially infected or may not remain infected by the time of their second blood meal. In such instances, the *Leishmania* parasite burden may not increase sufficiently for transmission despite the retroleptomonad-dependent population boost.

Accordingly, skin homogeneity has a particularly reduced role in very long lived sand flies that bite many times. In these flies, the number of metacyclics are repeatedly amplified, resulting in almost guaranteed parasite transmission to mammalian hosts at the third and subsequent blood meals for the majority, rendering such sand flies potential "super spreaders". To assess the impact of such flies, let us restrict the lifespans of the simulated flies to 20 days (Fig 4A, and see S5 Fig for the binary threshold equivalent). Restricting the lifespan of the flies to 20 days appears to have minimal effect on the influence of skin homogeneity, though a reduced plateau in mean $R_0$ value is achieved. This impact is predominantly due to the abbreviated capacity for metacyclic-enhancing blood meals in female sand flies with reduced lifespans. It should be noted that with a mean inter-bite time of 6 days, it is not unlikely that a given individual could take 3 blood meals in 20 days.

We next consider a further restriction of the lifespan to 15 days (Fig 4B, and see S6 Fig for the binary threshold equivalent). Under this new, harsher restriction we see that skin

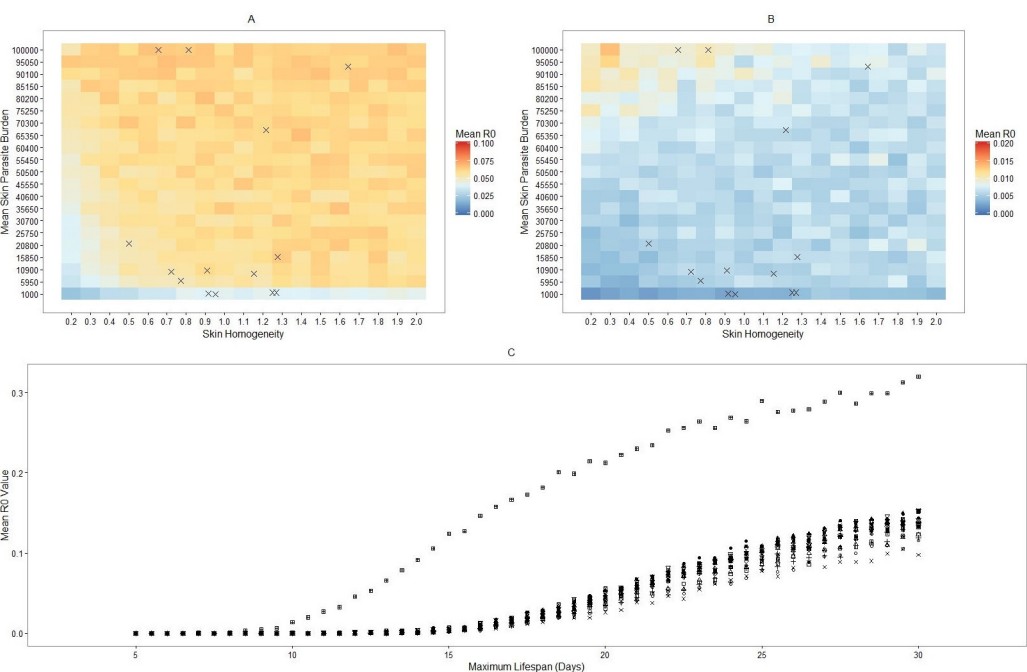

**Fig 4. Retroleptomonad dominance is dependent on having a sufficiently large maximum lifespan.** A, B) Heatmaps of the mean $R_0$ for simulated sand flies in Model B with 100% chance of biting an infected host and with lifespans restricted to 20 days (A) or 15 days (B). Crosses indicate the mean skin parasite burden and skin homogeneity of various mice from [7]. C) Mean $R_0$ value against maximum lifespan for RAG mice 1-18 from Doehl *et al.* [7] (S1 Table).

homogeneity has much stronger influence on the mean $R_0$ value. The peak observed in Model A is present again. The mean $R_0$ value does not drop to zero away from that peak, however. This is likely because some flies will still manage to bite three times and thus benefit from the retroleptomonad replicative cycle (this could also be interpreted as having a low, but non-zero, $\rho$ and thus we would expect a similarly low but non-zero mean $R_0$).

Further simulations based on the Doehl *et al*. mice help elucidate the transition between these two states. Using the parameterisation for mice 1-18 from Doehl *et al*. [7] (S1 Table), we ran sets of 5,000 sand flies for each mouse for a range of different maximum lifespans and calculated the mean $R_0$ value for each set. We can then compare the trajectory taken by the mean $R_0$ value for each population of simulated sand flies as we increase the maximum lifespan (Fig 4C).

We note that the mean $R_0$ value increases with the maximum sand fly lifespan for all mice, especially once it exceeds 15 days, as anticipated from Fig 4A and 4B. As sand fly longevity increases it stimulates a smooth transition away from a patchiness-dominated scenario and towards a retroleptomonad-dominated scenario. Thus the conclusions of Doehl *et al* [7] do not hold for flies with unrestricted lifespans, but provide valuable insight into the transmission potential of shorter-lived sand fly populations. Reducing the maximum lifespan of the sand flies (and thus enlarging the shorter-lived portion) can have a tangible impact on the mean $R_0$ value.

It is important to consider the sensitivity of our conclusions to certain model assumptions. Firstly, we have not fully addressed the effect of *Leishmania* infection on the sand fly vector. It has been documented that sand flies experience a reduction in their lifespan when infected [15], although the effect is not yet fully understood. In S3 Method, we modify the model to incorporate a 20% reduction in sand fly lifespan once infected. Supplementary S7 Fig demonstrates a quantitative reduction in mean $R_0$ but no qualitative changes to the behaviour of our system: we maintain the single peak exhibited by Model A, and the plateau of Model B. Though reduced, parasite infection and transmission dynamics are essentially unchanged.

We have also assumed that there exists a standard sand fly carrying capacity, suggesting a constant tolerance for infection by all parasite lifecycle stages. Supplementary S8B Fig shows the mean $R_0$ against maximum lifespan for a representative subsample of the RAG mice used by Doehl *et al*., as in Fig 4C, but in simulations where no limit to population size is imposed. We note that the results are almost indistinguishable from those of the full system (S8A Fig, Fig 4C). Our final sensitivity check removes the assumption of 100% efficiency in parasite differentiation. To represent this reduction in efficiency, we include a population sink at each lifecycle stage (see S3 Method for model specification and parameters). Supplementary S8C and S8D Fig correspond to the small and large sinks, respectively. Although Supplementary S8D Fig shows a marked decrease in mean $R_0$, in all cases we still observe the same qualitative relationship between mean $R_0$ and maximum lifespan.

## Discussion

We observe both numerically and analytically that the inclusion of retroleptomonads allows sand flies which take multiple bites to transfer more parasites on subsequent bites and thus be more effective at transmitting leishmaniasis, as anticipated by Serafim *et al* [11]. Less trivially, we also observe that the inclusion of retroleptomonad-dependent amplification in the model alters the relationship between the mean $R_0$ and skin homogeneity. In scenarios where the retroleptomonad life cycle stage is absent (Model A) or play a substantially reduced role (Fig 4B) we see a strong dependence on skin homogeneity, with patchy environments leading to more transmissions as some flies take up many parasites and can then cause infections, as predicted

by Doehl *et al* [7]. In scenarios where retroleptomonads are more important however, we see the opposite: skin homogeneity is unimportant to the transmission of the disease, as even small numbers of parasites initially present can be amplified greatly.

This result may reduce the perceived importance of the predictions made by Doehl et al. [7], yet there are important considerations that highlight its relevance. Doehl *et al.* predicted that patchy skin distributions would enhance transmissions because sand flies could occasionally take up higher parasite loads and then can lead to increased sand fly and subsequent mammal infections. Homogeneous skin environments, on the other hand, would reduce the likelihood of the *Leishmania* parasite establishing an initial sand fly infection. While we observe the loss of the relationship between skin homogeneity and mean $R_0$ for the full system there are scenarios where it re-emerges. Flies with short lifespans (Fig 4B) cause more transmissions with patchy than even skin distributions. Such sand flies are unlikely to live long enough to bite three or more times and thus the parasite populations do not typically benefit from the amplification step of the retroleptomonad stage in the model. This is reflected in our analyses. Consider the short-lifespan flies to have a low chance of taking three bites (IE a low $\rho$), then from Eq 22 we see that low k values increase the chance of transmission. Thus, there are conditions under which the scenario posed by Doehl *et al.* is relevant to the spread of the parasite. Perhaps an important caveat to the *in vivo* infection study is that immunodeficient mice from Doehl *et al.* may not properly represent a typical immunocompetent individual. While patchiness has not be reported in immunocompetent mice, the phenomenon of patchy skin parasite distributions remains applicable to clinically symptomatic Post-Kala Azar Dermal Leishmaniasis (PKDL) patients.

The extent to which our model's outcomes apply to parasite transmission in natural settings is uncertain. Multiple lab-based studies suggest that female sand flies have fairly short adult lifespans (<20 days) [16] with further reductions when infected [15]. Lab-based sand fly viability estimates are confounded by numerous challenges in maintaining sand fly colonies [17] and additional mortality associated with factors such as oviposition [18] and bacterial infection [19] that do not appear to impact wild populations as prominently. Release-recapture studies in natural settings suggest that flies may live much longer than in lab environments [20]. To address this uncertainty, we have incorporated parasite-induced mortality for an exemplar scenario to begin to assess its influence upon *Leishmania* transmission. Though this new addition did not alter the qualitative behaviour of this system for our exemplar scenario, we did observe a reduction in mean $R_0$ in all tested parameter combinations. This mean $R_0$ reduction will grow in magnitude for more severe lifespan reductions. We would also observe a loss of the plateau in Model B if the parasite-induced mortality was sufficiently severe to prevent the retroleptomonads from emerging. Such scenarios are, however, unlikely to be reasonable. In order to properly model the impact of parasite-induced mortality on the transmission potential of sand flies, it will be crucial for future studies to discern the true expected lifespan of wild sand flies and the full extent to which this lifespan is reduced by *Leishmania* parasite infection.

Transmission dynamics are further complicated by the feeding behaviour of the sand flies. We chose to model the time between subsequent blood meals (in days) using a gamma distribution of mean 6. Though this is a reasonable approximation for our model, in reality there is little information available about how often sand flies feed. It is likely that the feeding rate is linked to the oviposition cycle (given the dependence of oviposition on a blood meal) and the abundance of potential blood sources and promiscuous feeding behaviour exhibited by sand flies [6]. The scenario of regular feeds posed by Serafim *et al* [11] is a significant improvement upon theories which incorporate only a second blood meal at day 12. This seems appropriate for sand flies with abundant sources of blood meals, yet it is not uniformly true for all populations. We also consider human populations with different proportions of initially infected

hosts ($P_i$) including values such as 25% and 10% which are more applicable to populations where leishmaniasis is endemic [21, 22]. Although we observe that our results hold for such scenarios, we assume that hosts are evenly distributed throughout the populations and this is unlikely to be biologically accurate.

There is significant evidence that the behaviour of the sand flies is also altered once infected. A notable component of *Leishmania* infection known to alter sand fly behaviour is Promastigote Secretory Gel (PSG), a filamentous proteophosphoglycan-based gel secreted into the thoracic midgut and stomodeal valve [2, 3]. The occupation of the midgut by PSG causes the sand flies to feed ineffectively, taking smaller blood meals [3, 23] and demonstrating increased persistence when disturbed (with an increased likelihood of biting a second host after a disturbance) [15]. PSG also acts as a filter allowing only metacyclics to pass through [3], and impedes the unidirectional flow of blood through the stomodeal valve, causing the sand fly to regurgitate PSG and the parasites within it into the bite. This may amplify the number of infectious parasites transferred to a new host on a successful bite [3, 24]. Giraud *et al.* [25] recently investigated the complexity of this impact upon transmission. They reported that sand flies could regurgitate high "quality" (metacyclic-enriched) parasite doses even after multiple successive bites in a feed, likely due to PSG acting as a filter [3], but subsequent maintenance varies as the infection progresses in the fly. They also report that differences in dose quality have tangible impacts on the trajectory of the resulting infection in a mouse host, with lower quality bites often leading to larger, but less outwardly infectious lesions.

The interactions between PSG, fly feeding behaviour, and *Leishmania* population dynamics could have important implications for transmission. Sand flies that do feed on multiple hosts during a feed [15] could cause multiple infections given the enriched doses they may transmit, and the variable dose quality [25] may contribute to the emergence of variable patchiness in the skin of mammalian hosts observed by Doehl *et al* [7]. Although we model the regurgitation of parasites by increasing the number of transferred metacyclics for heavily infected flies [26], we do not directly model the PSG due to insufficient information regarding its production and how it interacts with the parasites in the midgut. Similarly the role of superspreading in *Leishmania* transmission, though beyond the scope of this study, may have significant implications for future models.

Another avenue of future enquiry that holds potential value relates to improving the parameterisation of our model. As the discovery of the retroleptomonad lifecycle stage is very recent [11] we have insufficient data to parameterise Model B with accuracy. Although our chosen parameters are informed by the population graphs of Serafim *et al.* and we can demonstrate that our model produces similar behaviour to that of the experimental system, it would be preferable to have more data to base our parameters upon. Future studies may seek to improve the identification of retroleptomonads using transcriptomics tools as has been done for previous life cycle stages [27]. Alternatively, they may seek to provide more information about the two lifecycle stages we omit from our model, the amastigotes and procyclic promastigotes. Either of these options would greatly improve predictions from future models.

## Conclusion

This work has produced a basic population dynamic model for nectomonad, leptomonad and metacyclic promastigotes and integrated the recently discovered retroleptomonad promastigote. This model can be further enhanced via the addition of missing life cycle stages or additional parameter to improve the fit. This provides a basic tool that can be expanded upon depending on the aims of a study. For example, a similar model may prove useful if modelling the impact of interventions on promastigote dynamics. Through using Monte Carlo

Simulations, we have demonstrated that the addition of retroleptomonads to the model greatly enhances transmission from the second bite onwards. This could suggest that retroleptomonads are a good stage to target in control efforts, potentially through interventions that reduce the number of bites a sand fly takes. We have also demonstrated that skin parasite heterogeneity does have an impact on *Leishmania* transmission, although a much smaller impact than retroleptomonads. A patchy distribution slightly enhances transmission when retroleptomonads are not present (such as the first bite), but a non-patchy distribution enhances transmission when retroleptomonads develop.

## Materials and methods

Model parameterisation was performed in RStudio v1.2.5019 (R version 3.6.1) with the digitize package [28] using data from [3] (see Supplementary S1 Method for full details). All Monte Carlo simulations were performed in MATLAB R2019b. Data analysis was performed in RStudio v1.2.5019 (R version 3.6.1).

## Supporting information

**S1 Table. RAG Mouse parameter combinations.** The skin heterogeneity and mean skin parasite burden values for RAG mice 1-18 used throughout our simulations, as originally calculated by Doehl *et al.* [A]: Values derived from Doehl *et al.* [7].
(PDF)

**S1 Method. Parameterisation of Model A.**
(PDF)

**S2 Method. Bite mechanics.**
(PDF)

**S3 Method. Population sink mechanics.**
(PDF)

**S1 Code. Supplementary code.** All MATLAB and R code comprising our implementation of the models and simulations used in this investigation.
(7Z)

**S1 Fig. Replicating the results of [11] (parasite proportions).** Comparison of the proportions of metacyclics (top) and retroleptomonads (bottom) at specific days throughout the lifespan of the simulated flies. Blue represents flies that bite only at day 0, orange represents flies that bite at day 12. The two categories are combined prior to day 12.
(TIF)

**S2 Fig. Evaluating model robustness by randomising parameters.** Number of metacyclics within the sand flies at specific days, with all parameters randomised prior to the start of each simulation. Parameters lie within 10% of the default value (Table 1). Blue represents flies that bite only a day 0, orange represents flies that bite at day 12.
(TIF)

**S3 Fig. Additional infected host proportions reflect the retroleptomonad dominance.** Heatmaps of the Mean $R_0$ for simulated sand flies for both Model A (left half) and B (right half) with 100% (top row), 50% (second row), 25% (third row), and 10% (bottom row) chance of biting an infected host, with the smooth transmission threshold function.
(TIF)

**S4 Fig. Heatmap dynamics remain qualitatively similar under a binary transmission threshold.** Heatmaps of the Mean $R_0$ for simulated sand flies for both Model A (left half) and B (right half) with 100% (top row), 50% (second row), 25% (third row), and 10% (bottom row) chance of biting an infected host, with the binary transmission threshold.
(TIF)

**S5 Fig. Reduced lifespan (20 days) dynamics remain qualitatively similar under a binary transmission threshold.** Heatmap of the Mean $R_0$ for simulated sand flies in Model B with 100% chance of biting an infected host and with lifespans restricted to 20 days, with the binary transmission threshold.
(TIF)

**S6 Fig. Reduced lifespan (15 days) dynamics remain qualitatively similar under a binary transmission threshold.** Heatmap of the Mean $R_0$ for simulated sand flies in Model B with 100% chance of biting an infected host and with lifespans restricted to 15 days, with the binary transmission threshold.
(TIF)

**S7 Fig. The inclusion of parasite induced mortality results in quantitative, but not qualitative, changes.** Heatmaps of Mean $R_0$ for simulated sand flies for both Model A (left half) and B (right half) with 100% (top row) or 25% (bottom row) chance of biting an infected host, with a smooth transmission threshold. After infection, sand flies receive a 20% reduction to their remaining lifespan.
(TIF)

**S8 Fig. Removing crucial assumptions of the model has minimal influence.** Mean $R_0$ against maximum lifespan for a representative subsample of RAG mice. A) Full model adapted from Fig 4c. B) Full model, but with no carrying capacity. C) Full model, but with additional small population sinks. D) Full model, but with larger population sinks.
(TIF)

## Author Contributions

**Conceptualization:** Samuel Carmichael, Ben Powell, Pegine B. Walrad, Jonathan W. Pitchford.

**Data curation:** Jonathan W. Pitchford.

**Formal analysis:** Samuel Carmichael, Ben Powell, Thomas Hoare, Pegine B. Walrad, Jonathan W. Pitchford.

**Funding acquisition:** Pegine B. Walrad, Jonathan W. Pitchford.

**Investigation:** Samuel Carmichael, Thomas Hoare.

**Methodology:** Samuel Carmichael, Pegine B. Walrad, Jonathan W. Pitchford.

**Project administration:** Jonathan W. Pitchford.

**Resources:** Pegine B. Walrad, Jonathan W. Pitchford.

**Software:** Samuel Carmichael, Thomas Hoare, Jonathan W. Pitchford.

**Supervision:** Ben Powell, Pegine B. Walrad, Jonathan W. Pitchford.

**Validation:** Jonathan W. Pitchford.

**Visualization:** Samuel Carmichael.

**Writing – original draft:** Samuel Carmichael, Ben Powell, Thomas Hoare, Jonathan W. Pitchford.

**Writing – review & editing:** Samuel Carmichael, Ben Powell, Pegine B. Walrad, Jonathan W. Pitchford.

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
