## [Decision Letter · Decision Letter 0]

30 Sep 2020

Dear Mr. Carmichael,

Thank you very much for submitting your manuscript "Variable bites and dynamic poplations; new insights in Leishmania transmission." for consideration at PLOS Neglected Tropical Diseases. As with all papers reviewed by the journal, your manuscript was reviewed by members of the editorial board and by several independent reviewers. In light of the reviews (below this email), we would like to invite the resubmission of a significantly-revised version that takes into account the reviewers' comments. 

We cannot make any decision about publication until we have seen the revised manuscript and your response to the reviewers' comments. Your revised manuscript is also likely to be sent to reviewers for further evaluation.

Sincerely,

José M. C. Ribeiro

Associate Editor

Jesus Valenzuela

Deputy Editor

Reviewer's Responses to Questions

**Key Review Criteria Required for Acceptance?**

**Methods**

-Are the objectives of the study clearly articulated with a clear testable hypothesis stated?

-Is the study design appropriate to address the stated objectives?

-Is the population clearly described and appropriate for the hypothesis being tested?

-Is the sample size sufficient to ensure adequate power to address the hypothesis being tested?

-Were correct statistical analysis used to support conclusions?

-Are there concerns about ethical or regulatory requirements being met?

Reviewer #1: The objectives of the study are clear. The authors develop a mathematical model using experimental data based on two recently published papers. Please see attached review report.

Reviewer #2: Yes, the model is clearly formulated

Reviewer #3: Yes

**Results**

-Does the analysis presented match the analysis plan?

-Are the results clearly and completely presented?

-Are the figures (Tables, Images) of sufficient quality for clarity?

Reviewer #1: Please see attached review report.

Reviewer #2: Yes, my response refers to the algebraic steps regarding the modeling approach.

Reviewer #3: Yes

**Conclusions**

-Are the conclusions supported by the data presented?

-Are the limitations of analysis clearly described?

-Do the authors discuss how these data can be helpful to advance our understanding of the topic under study?

-Is public health relevance addressed?

Reviewer #1: Please see attached review report

Reviewer #2: Yes, the conclusions follow the modeling formulation.

Reviewer #3: Yes

**Editorial and Data Presentation Modifications?**

Reviewer #1: Please see attached review report.

Reviewer #2: (No Response)

Reviewer #3: None

**Summary and General Comments**

Reviewer #1: Please see attached review report.

Reviewer #2: See attached.

Reviewer #3: Re. Carmichael et al.

The authors present a mathematical model of Leishmania transmission integrating two recent major findings in this field, built upon data acquired from another study into Leishmania infection in sand flies. They find that the distribution of amastigotes across the skin available for pick up, will have more of an influence on parasite transmission in short lived flies, whereas the presence of retroleptmonads and reamplification of the pool of metacyclic promastigotes in response to a second bloodmeal has a greater effect in longer lived flies.

I am not a mathematical expert, so I cannot comment on that side of this work, but I can offer insight into the Leishmania-sand fly interaction. 

On the whole I found this a really well designed and written piece of work, which brings fresh insight to the studies on which it is based. Below are the main comments I have: 

1. Clarity is needed on how you derived the retroleptomonad differentiation and replication rates and metacyclic differentiation rates? You quote personal communication with a co-author but this is needs to be explained in greater detail. It is also a discussion point as it raises gaps in our current knowledge.

2. The authors should consider incorporating the findings of Giruad et al. 2019, Communications Biology, into their discussion. They found that heterogeneity in the quality of the transmission (% metacyclics and non-metacyclics deposited) impacts on the course of the infection and transmission back to sand flies. This introduces the possibility of some sand flies acting as super-spreaders. Interestingly, the quality of the transmitted dose can also vary as the (primary) infections mature in the sand fly. This study did not look at the transmission dynamics from second bloodmeal fed flies.

3. Finally, as the interpretation of the author’s model has a lot to do with the lifespan of the infected vector, the authors should incorporate the role of parasite-induced mortality into it. In the discussion, the authors highlight that there is uncertainty in this area because there is a lack of evidence from wild populations. By extension, this argument could apply to all the data within this model. Surely, if sand fly longevity is an important to understand the implications of this model then it should form part of it, particularly, as there is sufficient data in the literature to do this. If this is not possible, the author’s should speculate on how this will influence their model in the discussion.

In conclusion, the study is excellent – it is impactful and offers fresh insight into the complex interaction between parasite, vector and host and is able to make important points relevant to the control of leishmaniasis. I believe this work is worthy of publication in PLoSNTD, if parasite-induced mortality forms a part of their model.

PLOS authors have the option to publish the peer review history of their article (what does this mean?). If published, this will include your full peer review and any attached files.

Reviewer #1: No

Reviewer #2: No

Reviewer #3: No
---

## [Editor Report · Decision Letter 1]

2 Dec 2020

Dear Mr. Carmichael,

We are pleased to inform you that your manuscript 'Variable bites and dynamic poplations; new insights in Leishmania transmission.' has been provisionally accepted for publication in PLOS Neglected Tropical Diseases.

Best regards,

José M. C. Ribeiro

Associate Editor

Jesus Valenzuela

Deputy Editor

---

## [Editor Report · Acceptance letter]

19 Jan 2021

Dear Mr. Carmichael,

We are delighted to inform you that your manuscript, "Variable bites and dynamic populations; new insights in Leishmania transmission.," has been formally accepted for publication in PLOS Neglected Tropical Diseases.

Best regards,

Shaden Kamhawi

co-Editor-in-Chief

Paul Brindley

co-Editor-in-Chief
